# Initial-site characterization of hydrogen migration following strong-field double-ionization of ethanol

Travis Severt[1], Eleanor Weckwerth [2], Balram Kaderiya[1], Peyman Feizollah [1], Bethany Jochim [1], Kurtis Borne[1], Farzaneh Ziaee[1], Kanaka Raju P [1,4], Kevin D. Carnes [1], Marcos Dantus[3], Daniel Rolles [1], Artem Rudenko [1], Eric Wells [2] ✉ & Itzik Ben-Itzhak [1] ✉

An essential problem in photochemistry is understanding the coupling of electronic and nuclear dynamics in molecules, which manifests in processes such as hydrogen migration. Measurements of hydrogen migration in molecules that have more than two equivalent hydrogen sites, however, produce data that is difficult to compare with calculations because the initial hydrogen site is unknown. We demonstrate that coincidence ion-imaging measurements of a few deuterium-tagged isotopologues of ethanol can determine the contribution of each initial-site composition to hydrogen-rich fragments following strong-field double ionization. These site-specific probabilities produce benchmarks for calculations and answer outstanding questions about photo-fragmentation of ethanol dications; e.g., establishing that the central two hydrogen atoms are 15 times more likely to abstract the hydroxyl proton than a methyl-group proton to form $H_3^+$ and that hydrogen scrambling, involving the exchange of hydrogen between different sites, is important in $H_2O^+$ formation. The technique extends to dynamic variables and could, in principle, be applied to larger non-cyclic hydrocarbons.

The migration of hydrogen within polyatomic molecules, either collectively[1–3] or as a single atom or ion at a time[4–10], is a common process that impacts diverse applications such as enzyme operation[11,12], large-scale studies of proteins[13], combustion[14–16], and atmospheric chemistry[17]. Roaming is a type of migration that bypasses the minimum energy pathway, usually through extended geometries and with timescales of several vibrational periods or more[1,2,6,9,18]. Despite the fact that many of these applications appear predominantly in neutral molecules, most recent gas-phase experiments use ions to probe the dynamics of hydrogen migration mechanisms, including roaming[1,2,9,10,19–27]. Ion-based experiments are favored in these situations because they generally have more available observables. The

experiments are often supported by molecular dynamics calculations of various types[28–34].

Comparisons between theoretical calculations and measurements of ionic molecular fragments are important because they benchmark the same theoretical approaches that are applied to the neutral molecules in which many applications occur. Photo-induced intra-molecular dynamics are usually explored theoretically by combining high-level electronic structure calculations with molecular dynamics methods (e.g., refs. 1,3,9,10,23,25). The associated fundamental questions about the coupling of electronic and nuclear dynamics are applicable to both ionic and neutral species, and roaming is known to occur in many situations[2,35,36].

[1]J. R. Macdonald Laboratory, Physics Department, Kansas State University, Manhattan, KS 66506, USA. [2]Department of Physics, Augustana University, Sioux Falls, SD 57108, USA. [3]Department of Chemistry, Michigan State University, East Lansing, MI 48824, USA. [4]Present address: School of Quantum Technology, DIAT (DU), Pune, Maharashtra 411025, India. ✉e-mail: eric.wells@augie.edu; ibi@phys.ksu.edu

While hydrogen atoms may be distinguished in calculations, they are fundamentally indistinguishable in experiments, thus complicating experimental to theoretical comparisons when the parent molecule has multiple non-equivalent hydrogen sites. Deuterium tagging, the targeted replacement of a hydrogen atom with a deuterium atom, has been a beneficial approach in many applications[37–40]. Of special relevance to the present work are mass-spectrometry and ion-imaging studies in which quantitative comparisons of different tri-hydrogen ($H_3^+$) formation processes are obtained in ethane[41,42] and methanol[1,43,44]. Expanding this work beyond molecules with two nonequivalent hydrogen sites is a challenge since there are only two readily available isotopes of hydrogen, and thus no single deuterium-tagged experiment can identify the role of all hydrogen sites.

The goal of our study is to extend deuterium tagging to determine the contribution of each hydrogen site in larger molecules, which display compelling hydrogen migration dynamics. This will allow quantitative comparisons that will fortify the connection between experiment and theory. Ethanol is an excellent system for this task, because it has three hydrogen sites and has attracted considerable experimental and theoretical interest due to the complexity of its dissociation[2,7,23,25–27,45–48].

In this article, we demonstrate that a combination of ion coincidence momentum-imaging (COLTRIMS)[49–51] measurements, using a few deuterium-tagged isotopologues of ethanol, can determine from which molecular sites the hydrogen atoms that compose a molecular fragment originated. The resulting site-specific probabilities for the ion composition provide an important new benchmark for molecular dynamics calculations. Instead of qualitative comparisons between experimental results and molecular dynamics calculations, the technique presented here makes possible quantitative comparisons of the likelihood that hydrogen atoms from different initial sites contribute to a final dissociation product.

These comprehensive site-specific results yield significant additional insights into hydrogen migration and bond rearrangement in the dissociation of ethanol dications, resolving questions from previous work[2,7,23,25,46–48,52]. For example, tri-hydrogen ion, $H_3^+$, formation has been attributed to roaming of a $H_2$ moiety that abstracts a proton to create the final product[2,25], a pathway that was previously identified

in methanol[1]. In ethanol, the roaming $H_2$ is understood to be composed of the two central $\alpha$-hydrogen atoms [See Fig. 1a], but until now, the likelihood of abstracting the proton from the methyl- or hydroxyl-group has not been experimentally quantified. Our results show that abstraction of the hydroxyl hydrogen is far more probable, although every possible $H_3^+$ initial-site composition occurs. As shown in Fig. 1b, double-hydrogen migration to the hydroxyl group has been identified as a likely formation mechanism for the hydronium ion[23], $H_3O^+$. We measure a nearly statistical ratio between the possible initial-site compositions yielding $H_3O^+$ that can result from double-hydrogen migration to the hydroxyl group, which suggests that the two hydrogen atoms migrate independently, consistent with previous time-resolved findings[23].

The complete analysis results in site-specific probabilities for the formation of tri-hydrogen ($H_3^+ + C_2H_3O^+$), hydronium ($H_3O^+ + C_2H_3^+$), water ($H_2O^+ + C_2H_4^+$), and methane ($CH_4^+ + CH_2O^+$) ions following double ionization of ethanol by an intense ultrafast laser pulse. Characterization of all of the possible resulting ion compositions shows that seemingly unlikely processes, such as hydrogen scrambling[41,42,53,54], are sometimes significant.

Furthermore, for $H_3^+$ formation, we extend the analysis to include not only results of two-body breakup of the dication, namely $H_3^+ + C_2H_3O^+$, but also cases when the dissociation of the ethanol dication includes the elimination of one or two hydrogen atoms. In these latter cases, the site-specific conditional probability for both the hydrogen composing the $H_3^+$ and the individual eliminated hydrogen atoms is resolved. The ability to reconstruct the net momentum of the neutral fragments is one of the major experimental advantages of the coincidence ion momentum imaging technique. We observe that concurrent elimination of a hydrogen atom can alter the relative probabilities of the different $H_3^+$ initial-site compositions substantially.

While this experiment was done using ethanol as a prototype system, we show that the method can be extended to larger non-cyclic molecules composed of carbon, hydrogen, and oxygen. Moreover, the process can be applied to other dynamical variables, such as the kinetic energy release (KER), which provides the opportunity to further evaluate molecular dynamics calculations and deepen our understanding of the dynamics at play.

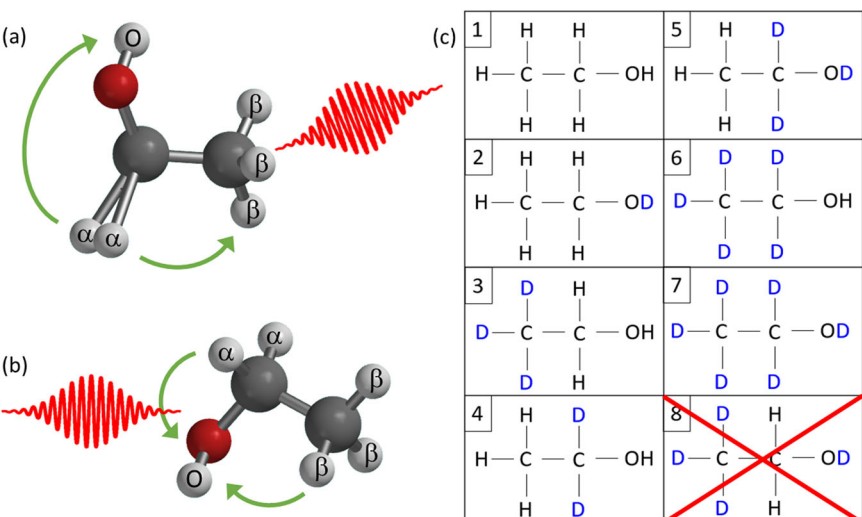

**Fig. 1 | Ethanol schematic structure, breakup mechanisms, and isotopologues. a, b** Ball-and-stick cartoons of the ethanol molecule with the hydrogen atoms labeled ($\alpha$, $\beta$, or O) in the manner used in the rest of the article. **a** An illustration of the theoretically favored routes to $H_3^+$ formation as described by ref. 2 in which the $H_\alpha H_\alpha$ complex stretches and eventually roams in either direction to form $H_\alpha H_\alpha H_O^+$ or $H_\alpha H_\alpha H_\beta^+$. **b** The preferred theoretical pathway for $H_3O^+$ formation identified by

ref. 23, in which migration of the $H_\beta$ and $H_\alpha$ toward the hydroxyl group leads to stretching of the C-O bond and dissociation of the hydronium ion. **c** Specific isotopologues of ethanol, arbitrarily numbered for convenience. The first seven isotopologues were used in the experiment. The eighth isotopologue, $CD_3CH_2OD$, was not readily available at the time of the experiment, and thus, measurements were not performed on it.

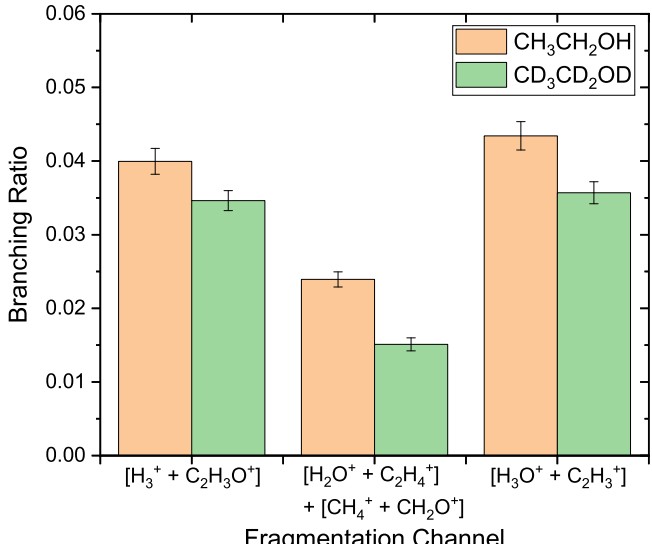

**Fig. 2 | Isotopic dependence of ethanol dication two-body $H_3^+$, $H_2O^+$, $CH_4^+$ and $H_3O^+$ branching ratios.** Comparison between the branching ratios, defined as the yield of a specific channel relative to all channels (See Eq. (1) in the Methods Section) of the two-body breakup channels of interest (marked explicitly on the figure), produced from the pure hydrogen and the pure deuterium isotopologues [#1 and #7 as denoted in Fig. 1c]. Note that the water and methane channels are combined because $D_2O^+$ and $CD_4^+$ cannot be distinguished from each other in this measurement. The error bars are the combination of the systematic and statistical uncertainties (see details in Supplemental Note 8). The isotopic differences are larger than the experimental uncertainty for some channels, but (as detailed in Supplemental Note 13) using the collection of all seven isotopologues reduces bias in the site-specific probabilities to a level that is characterized by the uncertainty in the results (see Figs. 3 and 4). Source data are provided as a Source Data file.

## Results

Ethanol, $CH_3CH_2OH$, has three non-equivalent sites containing hydrogen atoms, as shown in Fig. 1, specifically the hydrogen that is part of the hydroxyl group, the three $\beta$-hydrogen atoms in the methyl group, and the two $\alpha$-hydrogen atoms attached to the central carbon atom. Throughout this article, we refer to these hydrogen-atom sites as $H_O$, $H_\beta$, and $H_\alpha$, respectively.

The experimental strategy is to conduct COLTRIMS measurements of different isotopologues of ethanol under similar conditions. There are eight possible isotopologues of ethanol that have deuterium completely substituted for one or more of the $H_O$, $H_\beta$, or $H_\alpha$ sites, as illustrated in Fig. 1c. Measuring seven isotopologues was sufficient for the site-specific analysis that follows, although the eighth isotopologue ($CD_3CH_2OD$) would have provided useful additional data. All the ethanol isotopologues (1–7) were ionized by 23 fs FWHM laser pulses with a peak intensity of $3.0 \times 10^{14}$ W cm$^{-2}$ and a central wavelength of 790 nm. The three-dimensional momenta of the resulting ions were measured in coincidence. Keeping the experimental conditions the same across all measurements was a point of emphasis, as detailed in the Methods section.

As with many deuterium substitution techniques, the fundamental assumption is that the mass difference between $^1H$ and $^2D$ does not significantly affect the properties to be studied. This common assumption[41,42,55,56] was verified in our case by comparing the branching ratios from the all-hydrogen isotopologue, #1 in Fig. 1c, to the analogous channels from the all-deuterium isotopologue, #7 in Fig. 1c. The all-H and all-D branching ratios are quite similar in magnitude and resemble the size of mass effects previously observed from other small molecules[57,58]. As shown in Fig. 2, the mass dependence of the bond rearrangement channels is larger than the experimental uncertainty, but it is smaller than the observed difference in methanol[1], and the

effects tend to cancel in our site-specific analysis. This is discussed in more detail in Supplementary Note 13 (SN 13).

Narrow diagonal stripes in the coincidence-time-of-flight (CTOF) plot identify different breakup channels of the ethanol dication [see Figs. 2–3 in SN 8]. The yield in each channel is used to define a fragmentation branching ratio, as described in the Methods section. While the branching ratios from any single isotopologue cannot identify the likelihood of all possible initial-site compositions yielding the fragment ions of interest, each isotopologue provides partial composition information. As detailed in the Methods section, however, the combination of all seven measured isotopologues overdetermines the site-specific probabilities. In the case of the $H_3^+ + C_2H_3O^+$ channel, for example, 10 measured branching ratios specify six possible hydrogen compositions. A least-squares fitting procedure optimizes the site-specific probabilities to the measured data.

Four final products are examined using the analysis: tri-hydrogen, hydronium, water, and methane ions. The results are summarized in Fig. 3 for complete breakup channels, while the incomplete fragmentation channels are presented in Fig. 4. In the context of the measured ion-pairs, complete and incomplete breakup channels are defined as two-body breakup and few-body (3 or more) breakup, respectively. Note that the few-body breakup of interest here involves the elimination of one or two hydrogen atoms. For complete breakup, $H_3O^+$ is the most likely of the four ions to be observed, although $H_3^+$ has a branching ratio that is 90% of the $H_3O^+$ value. $H_2O^+$ and $CH_4^+$ are somewhat less likely, with branching ratios of 28% and 21% of the $H_3O^+$ level, respectively.

In Fig. 3, we show violin plots[59] of the conditional probability for each initial-site composition of tri-hydrogen, hydronium, water, and methane ions formed in two-body breakup. The violin plots indicate the estimated uncertainty of the fits, as detailed in the Methods section. At a glance, one can identify the dominant initial-site compositions as well as those with negligible probability. The most likely initial-site composition for $H_3^+$ is $H_\alpha H_\alpha H_O$, for $H_3O^+$ it is $H_\beta H_\alpha H_O O$, for $H_2O^+$ it is $H_\beta H_O O$, and for $CH_4^+$, the methyl group is more likely to capture an alpha proton than a hydroxyl proton.

Similar violin plots, shown in Fig. 4, illustrate the conditional probability for the more complex few-body incomplete breakup channels producing a tri-hydrogen ion and either one or two neutral hydrogen atom(s). Furthermore, we show the conditional probability for specific initial sites from which the neutral hydrogen(s) is lost. For example, Fig. 4b shows that one-hydrogen elimination in $H_3^+$ formation from two $\beta$- and one $\alpha$-hydrogens is dominated by the elimination of the hydroxyl hydrogen. Figure 4b, d shows only a sample of the incomplete channels distinguished by the initial site(s) of the neutral hydrogen atom(s). A complete reporting is provided in SN 9 and SN 10.

In general, we determined comprehensive probabilities for the initial-site composition of hydrogen-rich ions in photofragmention of ethanol dications. The extension to larger molecules is, in principle, straightforward provided suitable isotopologues can be obtained. As molecules become larger, the number of possible distinct measurables often grows faster than the number of different sites that can contribute to the ion compositions. In SN 11, we explicitly prove that this is true for non-cyclic molecules composed of carbon, hydrogen, and oxygen, such as 1- and 2-propanol. Hydrogen migration and bond-rearrangement processes in these larger molecules have recently garnered attention[1,2,60,61]. The main experimental impediment to exploring these larger systems is the time needed to obtain data for all isotopologues, but laser systems running at hundreds of kilohertz[62] make these potential experiments more feasible.

Since the fragment momentum is calculated on an event-by-event basis, associated dynamic quantities can be calculated and binned. Provided there are sufficient statistics in each bin, the relative site-specific probability of a dynamic quantity can be calculated for each bin in the same manner as the relative probability of the composition

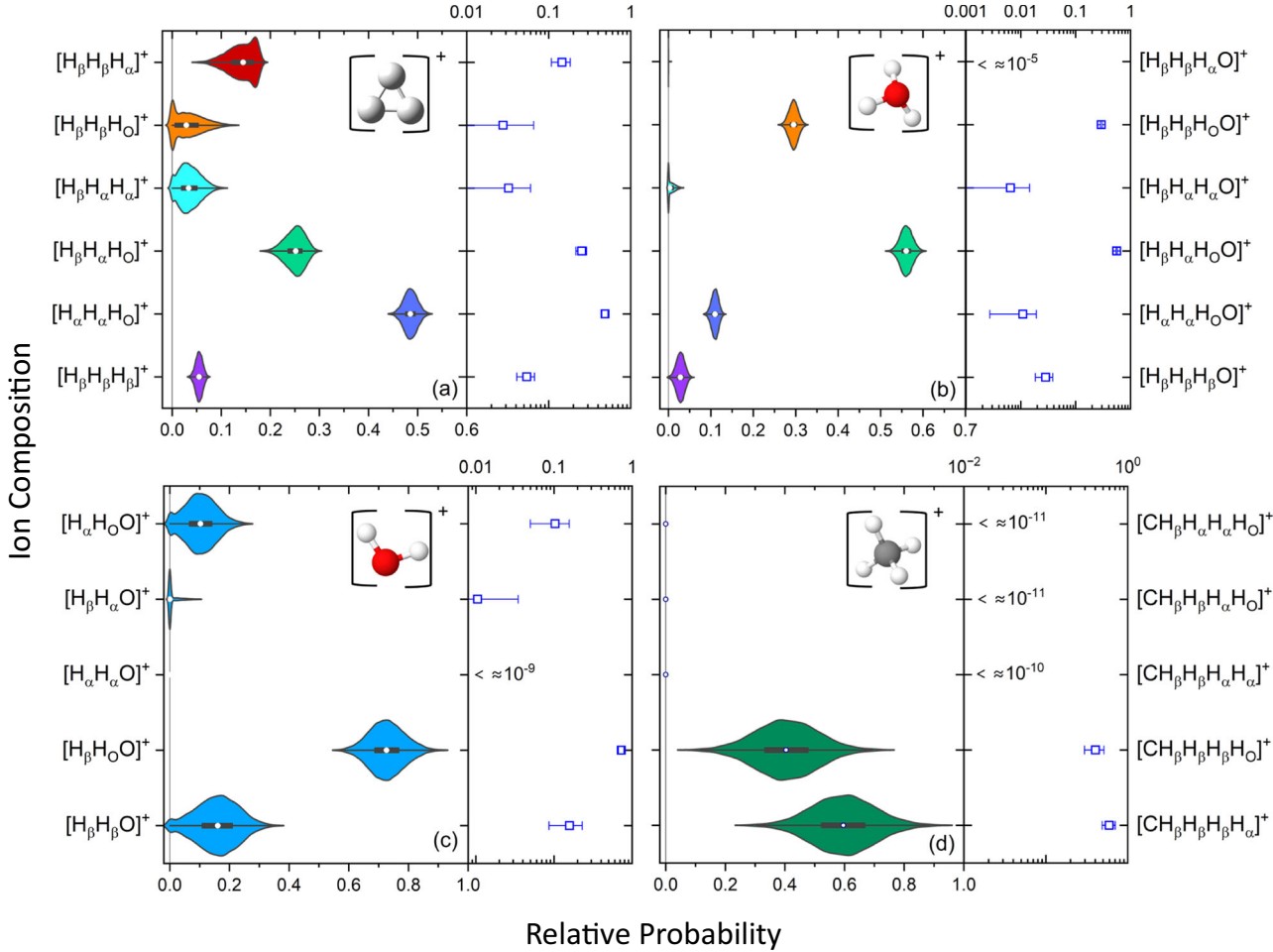

**Fig. 3 | Relative probabilities for different initial-site compositions of complete fragmentation channels of ethanol dications leading to $H_3^+$, $H_3O^+$, $H_2O^+$, and $CH_4^+$.** The left panel of each plot shows the probability distribution evaluated by Monte-Carlo uncertainty propagation as a violin plot, where the white circle is the median, the black rectangle indicates the middle 50% of the distribution, the thin black line indicates range of the distribution excluding outliers, and the (vertical) width of the distribution denotes the probability of each value. The right panel is a log-scale plot of the mean values of the same data with $1\sigma$ error bars, assuming normal statistics. **a** Results for $H_3^+ + C_2H_3O^+$. The overall probability for this channel, $P(H_3^+)$, is $0.0362 \pm 0.0020$. The relative probabilities shown in the panel, which sum to 1.0, are conditional so that the overall site-specific probability of $P(H_\beta H_\alpha H_O)$, for example, is the product of $P(H_3^+)$ and the value of $[H_\beta H_\alpha H_O]^+$ plotted in the panel. **b** The same format as (**a**) but now for $H_3O^+ + C_2H_3^+$ with $P(H_3O^+) = 0.03989 \pm 0.00098$. The same shading of the violin plots in (**a**) and (**b**) indicates the same initial site composition in $H_3^+$ and $H_3O^+$. **c** $H_2O^+ + C_2H_4O^+$ with $P(H_2O^+) = 0.0110 \pm 0.0014$. **d** $CH_4^+ + CH_2O^+$ with $P(CH_4^+) = 0.0083 \pm 0.0015$. All of the probabilities above are relative to complete fragmentation channels, that is, using only the first term in the denominator of Eq. (1), which is found in the Methods section. Source data are provided as a Source Data file.

of the entire fragmentation channel. For example, we determined the site-specific KER distribution of the $H_3^+ + C_2H_3O^+$ channel. The results, shown in Fig. 5, indicate that the different initial-site compositions are associated with different KER. While it is difficult to draw direct conclusions from these measured KER distributions without associated measurements of the photoelectron energies and knowledge of the potential energy surfaces, the site-specific KER can provide a useful consistency check for comparisons with theoretical predictions. For example, Wang and co-authors noted that the formation of vibrationally excited $H_3^+$ could account for the difference between the measured and calculated KER for the ethanol dication produced by electron impact ionization[25].

The fitting process used to determine the site-specific KER does not sacrifice the correlation between measurables that is a powerful feature of coincidence measurements, but the correlation does become limited by statistics. For example, if it was desired to determine the angular distribution associated with a particular range of the site-specific KER distribution, one would need enough statistics in the particular site-specific KER bin to distribute among the angular bins

and still produce an acceptable fit of the overdetermined set of equations for each angular bin. On the other hand, since the integrated KER distributions match the relative probabilities shown in Fig. 3a within about twice the experimental uncertainties despite being the result of ~20 independent fits, this process illustrates the robustness of the method.

## Discussion

We start our discussion with the complete two-body breakup channels for which the probabilities of each initial-site composition have been determined. Then, we examine the site(s) from which hydrogen atom(s) are eliminated during tri-hydrogen formation, thus extending the methodology to examine incomplete fragmentation channels.

### Complete breakup channels

Complete photofragmention channels of the ethanol dication are simpler to analyze due to the reduced number of possible fragmentation combinations, in contrast to photofragmentation that also produces a neutral fragment—discussed in the next section. Below, we

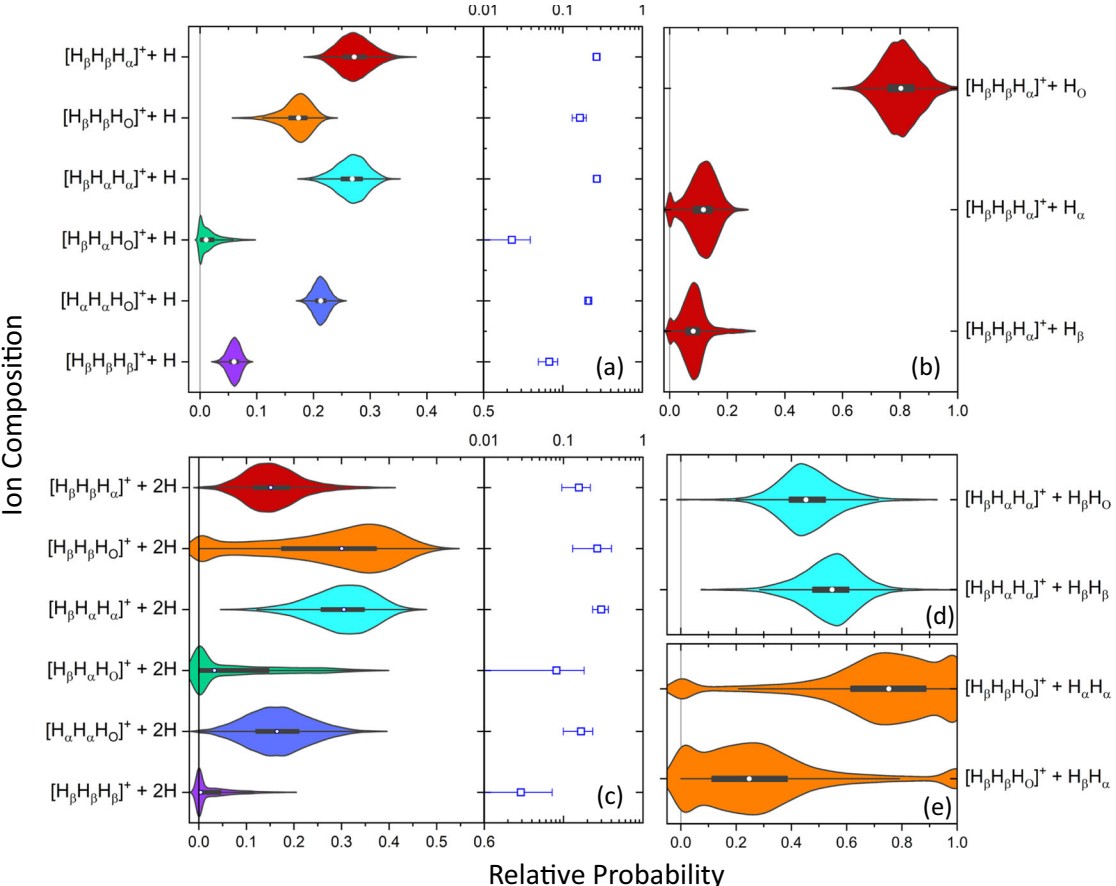

**Fig. 4 | Relative probabilities for incomplete fragmentation channels of ethanol dications leading to H$_3^+$. a** Violin plots (left) and log-scale plots of the mean (right) of the relative probabilities of different initial-site compositions for H$_3^+$ + C$_2$H$_2$O$^+$+ H with $P$(H$_3^+$,H) = (2.6 ± 0.2) × 10$^{-3}$. The 1$\sigma$ error bars on the log-scale plot assume a normal distribution. **b** Violin plots for the different hydrogen elimination sites associated with H$_\beta$H$_\beta$H$_\alpha$ (red) initial-site composition of the H$_3^+$ + C$_2$H$_2$O$^+$ + H channel, specifically H$_O$, H$_\alpha$, and H$_\beta$, with $P$(H$_\beta$H$_\beta$H$_\alpha$, H) = 0.267 ± 0.027. **c** Similar to (**a**) but for H$_3^+$ + C$_2$HO$^+$+ 2H, the incomplete breakup channel yielding two hydrogen

atoms (or one H$_2$ molecule) in addition to H$_3^+$ formation, with $P$(H$_3^+$,2H) = (3.6 ± 0.8) × 10$^{-4}$. **d, e** Violin plots for the hydrogen elimination sites associated with the two possible initial site compositions, (**d**) H$_\beta$H$_\alpha$H$_\alpha$ + C$_2$HO$^+$+ 2H (cyan) and (**e**) H$_\beta$H$_\beta$H$_O$ + C$_2$HO$^+$+ 2H (orange), with $P$(H$_\beta$H$_\alpha$H$_\alpha$,2H) = 0.299 ± 0.067 and $P$(H$_\beta$H$_\beta$H$_O$,2H) = 0.27 ± 0.14, respectively. The probabilities in this figure are derived using the full denominator of Eq. (1) in the Methods section. In all panels, the violin plot shading color is associated with the initial site composition of the H$_3^+$ fragment. Source data are provided as a Source Data file.

focus on the formation of tri-hydrogen, hydronium, water, and methane ions in two-body breakup of ethanol dications, in particular, the initial sites of the hydrogen atoms involved.

### Tri-hydrogen ion, H$_3^+$ + C$_2$H$_3$O$^+$

Formation of H$_3^+$ is the most often studied of the molecular ions mentioned above[1–3,7,19,22,24–27,44,52,63]. Figure 1a illustrates the primary initial step in the H$_3^+$ formation proposed by ref. 2, in which a neutral H$_2$ moiety forms and then abstracts an additional proton. Similar mechanisms have been suggested for methanol[1,3,24] and ethane[19]. Calculations described by Wang and co-workers[25] to model an experiment examining double-ionization of ethanol by electron impact also highlighted the importance of rapid (<50 fs and as low as 20 fs) H$_2$ formation in order to stabilize the dication, thereby allowing time for additional pathways to open before dissociation of the dication.

The previous reports[2,25] suggested that the H$_2$ formation, shown in Fig. 1a, occurs at the $\alpha$ site and leads, after roaming, to the tri-hydrogen ion in either H$_\alpha$H$_\alpha$H$_O$ or H$_\beta$H$_\alpha$H$_\alpha$ initial-site compositions. Neither of these reports, however, determined the relative importance of these two pathways. Our measurements show that abstracting a proton from the hydroxyl group is 15 times more likely than from the methyl group, i.e., H$_\alpha$H$_\alpha$H$_O$ is strongly favored over H$_\beta$H$_\alpha$H$_\alpha$. This strong preference is in spite of the fact that the latter has 3 possible permutations (i.e., multiplicity) while the former has only one. This result also contrasts

with the similar process in methanol, where the roaming H$_2$ is more likely to abstract the methyl proton than the hydroxyl proton[1,44].

While H$_\alpha$H$_\alpha$H$_O$ is the most likely initial-state composition, our results, shown in Fig. 3a, indicate that the H$_3^+$ + C$_2$H$_3$O$^+$ breakup channel occurs with all six possible compositions. The dominant H$_\alpha$H$_\alpha$H$_O$ composition occurs about half the time, followed by H$_\beta$H$_\alpha$H$_O$. The latter initial-site composition may be due to the independent migration of a $\beta$- and $\alpha$-site hydrogen to the hydroxyl group, a mechanism suggested by Kling et al. for hydronium formation[23], but in our case leading to tri-hydrogen ion formation instead. Ab initio simulations by Ekanayake et al.[2] showed that a roaming H$_\alpha$H$_\beta$ moiety could form, leading to H$_\beta$H$_\alpha$H$_O$. Wang et al.[25] suggested this H$_\alpha$H$_\beta$ moiety could be due to a more complex H$_2$ roaming mechanism, namely an $\alpha$ to $\beta$ migration that forms a transient CH$_4$ group, which releases a H$_2$ that roams from the $\beta$ site to the hydroxyl group. In this mechanism the H$_2$ is dislodged because of the initial hydrogen migration, and it may consist of either two $\beta$ hydrogens or a $\beta$- and $\alpha$-hydrogen pair, i.e., the latter is the hydrogen that migrated from the $\alpha$ site. Distinguishing between the double-migration and roaming H$_\beta$H$_\alpha$H$_O$ pathways is possible only in theoretical calculations. The roaming mechanism, however, should contribute to both measured H$_\beta$H$_\beta$H$_O$ and H$_\beta$H$_\alpha$H$_O$ site-specific probabilities. Our measurements indicate a strong preference for the H$_\beta$H$_\alpha$H$_O$ over the H$_\beta$H$_\beta$H$_O$ initial site compositions by almost an order of magnitude, a fact that tends to support the double-migration

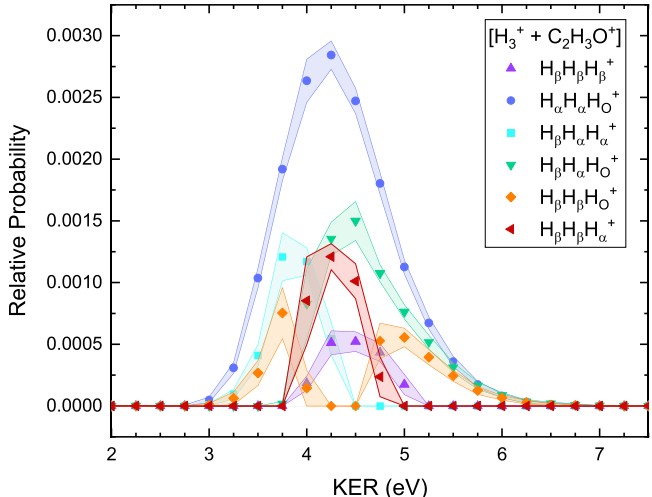

**Fig. 5 | Site-specific kinetic energy release (KER) distributions for $H_3^+$ formation in complete breakup channels.** The error bands (marked by the respective color shade) represent the range of the middle 50% of the Monte Carlo uncertainty propagation. The area under each curve is associated with the relative probabilities for that initial-site composition [i.e., the probabilities shown in Fig. 3a]. Source data are provided as a Source Data file.

mechanisms suggested by ref. 23, though calculations that provide the likelihood of each of the proposed mechanisms are needed to verify this assertion.

In a slightly different version of the process involving a transient $CH_4$ group, the emitted $H_2$ moiety was predicted to roam and then abstract a $\beta$-hydrogen (rather than the hydroxyl hydrogen as described in the previous paragraph)[25]. This process leads to $H_\beta H_\beta H_\beta$, which we observe to be twice as likely as the $H_\beta H_\beta H_O$ composition. The $H_\beta H_\beta H_\beta$ formation was also attributed to migration of an $\alpha$-hydrogen to the terminal methyl site, but in contrast to the mechanism discussed above, directly dislodges the $\beta$-hydrogens, which subsequently form a $H_3^{+}$ [2]. Finally, no mechanism leading to the third likely composition, namely $H_\beta H_\beta H_\alpha$, was suggested in refs. 2,25, though this composition has been previously identified by deuterium tagging[2]. In all these cases, the measured probability of each initial-site composition can encourage future theoretical work to identify the underlying mechanisms including a prediction of the probability of each proposed mechanism.

### Hydronium, $H_3O^+ + C_2H_3^+$
In $H_3O^+$ formation, it is very likely that the entire hydroxyl group is included in the final product. As shown in Fig. 3b, only 3.3% of all $H_3O^+$+ $C_2H_3^+$ channels do not include a $H_O$ atom in the $H_3O^+$. The most probable initial-site composition measured is $H_\beta H_\alpha H_O O$. This observation is consistent with the double-hydrogen migration mechanism suggested by ref. 23 and illustrated schematically in Fig. 1b. Their theoretical work indicated no preference between the migration from the initial $\alpha$ and $\beta$ sites, which is in agreement with our measured $(1.91 \pm 0.15)$:1 ratio between $H_\beta H_\alpha H_O O$ and $H_\beta H_\beta H_O O$, since the relative multiplicity would predict 2:1.

The most likely $H_\beta H_\alpha H_O O$ composition is also consistent with both of the main pathways identified by ref. 25 for $H_3O^+$ formation. These pathways involve hydrogen migration between the $\alpha$ and $\beta$ sites and thus either $H_\beta H_\alpha H_O O$ or $H_\alpha H_\alpha H_O O$ could result. We do observe $H_\alpha H_\alpha H_O O$, with a measured $H_\beta H_\beta H_O O$ to $H_\alpha H_\alpha H_O O$ ratio of $(2.66 \pm 0.35)$ : 1. The final permutation predicted by multiplicity is that the $H_\beta H_\alpha H_O O$ to $H_\alpha H_\alpha H_O O$ ratio is 6 : 1, and we measure $(5.1 \pm 0.59)$ : 1, a difference of $1.5\sigma$. Thus, it appears that when the entire hydroxyl group is included in the $H_3O^+$ product, the probability of the $H_\alpha$ or $H_\beta$ sites contributing to the product composition is approximately statistical,

which in turn supports the likelihood of independent migration of the hydrogen atoms. This result is compatible with the time-resolved findings of ref. 23, which showed no correlation between the migrating hydrogens.

The remaining hydronium initial-site composition with non-negligible probability is $H_\beta H_\beta H_\beta O$. This composition is an example of hydrogen scrambling[42,53,54], which in this case means that the $H_O$ must be displaced to the $C_2H_3^+$ fragment and the oxygen must bond with three other hydrogens. This scrambling process ultimately attaches all three $\beta$-hydrogens to the oxygen. Despite the fact that both the $H_\beta H_\beta H_\alpha O$ and $H_\beta H_\alpha H_\alpha O$ compositions have higher multiplicity than the $H_\beta H_\beta H_\beta O$ composition, the measured site-specific probability for both of them is consistent with zero within the experimental uncertainty.

### Water ion, $H_2O^+ + C_2H_4^+$
Hydrogen scrambling is also evident in the formation of water ions. As in the hydronium case, the only measurable composition of water ion without a $H_O$ is $H_\beta H_\beta O$, which is the second most probable water channel overall. The most likely $H_2O^+$ composition associated with $H_2O^+ + C_2H_4^+$ is $H_\beta H_O O$. Previous work[23,64] suggests $H_2O^+$ formation is the result of a single hydrogen migration. In their theoretical studies of $H_3O^+$ formation, Wang and co-workers found that roaming of $H_2O$ was a possible intermediate step prior to abstraction of a proton[25]. Presumably, a similar hydrogen migration from the $\beta$ site could initiate ejection of a $H_2O^+$ (instead of neutral water) in either the $H_\beta H_O O$ or $H_\alpha H_O O$ compositions, in agreement with single hydrogen migration pathways suggested by ref. 23 and theoretically supported more recently by ref. 64. The measured water ion compositions are summarized in Fig. 3c. We note that the measured ratio of $H_\beta H_O O$ to $H_\alpha H_O O$, $(7.0 \pm 3.7)$:1 is inconsistent with the 3:2 expectation from multiplicity. In contrast to the multiphoton and electron-impact results described above, the single (-100 eV) EUV photon-double ionization experiment conducted by ref. 26 only observed the formation of water ions through dissociation leading to more than two fragments.

### Methane ion, $CH_4^+ + CH_2O^+$
As shown in Fig. 3d, the measured composition of the methane ions in $CH_4^+ + CH_2O^+$ always includes all three $\beta$-hydrogens, i.e., no scrambling is observed. The fourth hydrogen is somewhat more likely to be $H_\alpha$ than $H_O$. The overall probability for $CH_4^+ + CH_2O^+$ dissociation is lower than the probability of the other complete coincidence channels described above. Wang et al.[25] point out that near the most probable Franck–Condon transition point, the field-free potential energy surface indicates that the $CH_4CHOH^{2+}$ configuration is stable against $C_\alpha$-$C_\beta$ bond breaking, which may explain why the overall probability of $CH_4^+ + CH_2O^+$ is small even though a transient $CH_4^+$ might be involved in several of the processes discussed above.

### Incomplete breakup channels involving $H_3^+$
When the fragments of the ethanol dication include a neutral particle (or particles), this is an indication that the system has additional internal energy and therefore different migration mechanisms can occur. This is evident in recent measurements with -100 eV photons, which produced $H_3O^+$ fragments from double-ionization of ethanol only by two-body dissociation and $H_2O^+$ fragments only in three-body dissocation[26]. Thus, a comparison between complete and incomplete breakup channels for $H_3^+$ in our study can give some insight into how the site-specific compositions change as higher excited states become accessible.

Experimentally, the determination of the yield of incomplete breakup channels from the CTOF data is less straightforward because the momentum carried by the undetected neutral fragment(s) leads to a broadening of the diagonal CTOF stripe (see Fig. 3a in SN 6). For this

reason, we restricted this portion of the analysis to channels that contain a $H_3^+$ fragment and one or two neutral hydrogen atoms because the tri-hydrogen ion channels are better separated on the CTOF plot than more massive ions. Some of the dissociation channels left out of the analysis are relatively large (e.g., $H_3^+ + CO^+ + CH_3$) due to the propensity for $C_\alpha$-$C_\beta$ bond breaking. The analysis of these incomplete channels with heavier neutral fragments is significantly more time consuming, although it is possible if there is sufficient interest. Figure 4 illustrates that the ejection of one or two neutral hydrogen atoms during the fragmentation of the ethanol molecule results in a widespread alteration of the relative $H_3^+$ site-specific probabilities compared to the complete breakup channels presented in Fig. 3. The two dominant initial-site compositions seen in $H_3^+ + C_2H_3O^+$, namely $H_\alpha H_\alpha H_O$ and $H_\beta H_\alpha H_O$, are significantly reduced in the $H_3^+ + C_2H_2O^+ + H$ channel [see Fig. 4a], with no statistically significant $H_\beta H_\alpha H_O$ observed. In contrast, the relative probabilities of the $H_\beta H_\beta H_\alpha$, $H_\beta H_\beta H_O$, and $H_\beta H_\alpha H_\alpha$ compositions increase.

When an additional hydrogen atom dissociates, the changes in relative composition between Fig. 4a, c are less dramatic than the differences between Fig. 3a and Fig. 4a. One noticeable change is that $H_\beta H_\beta H_\beta$ falls off substantially, from around 5% in both Figs. 3a and 4a to a value consistent with zero in Fig. 4c. It is possible that this occurs because it becomes unlikely to both disturb the methyl group and also eject two hydrogen atoms. The $H_\beta H_\beta H_O$ and $H_\beta H_\alpha H_\alpha$ compositions are the most likely to be associated with double hydrogen elimination while the $H_\beta H_\beta H_\alpha$ and $H_\alpha H_\alpha H_O$ compositions are about half as probable.

The overall increase in the range of outputs from the Monte Carlo uncertainty analysis as the number of ejected hydrogen atoms increases mostly reflects the drop in the total number of events and the associated increase in the statistical uncertainty. While the $H_3^+$ complete breakup channel is the same order of magnitude as the $H_3^+ + C_2H_2O^+ + H$ channel, the $H_3^+ + C_2HO^+ + 2H$ channel is approximately an order of magnitude less likely.

Figure 4b, d shows that each of the $H_3^+ + C_2H_2O^+ + H$ and $H_3^+ + C_2HO^+ + 2H$ channels can be further distinguished by the initial sites of the dissociating hydrogen atoms, not just by the hydrogen initial-site compositions yielding tri-hydrogen ions. In Fig. 4b, we show that the $H_\beta H_\beta H_\alpha$ composition of the $H_3^+ + C_2H_2O^+ + H$ channel is much more likely to be associated with a $H_O$ hydrogen elimination than either a $H_\alpha$ or $H_\beta$ hydrogen elimination. Figure 4d illustrates two cases of double hydrogen elimination, i.e., the $H_3^+ + C_2HO^+ + 2H$ channel. First, if tri-hydrogen is formed from the $H_\beta H_\alpha H_\alpha$ initial-site composition the two possible hydrogen-pair eliminations have roughly equal probabilities. In contrast, when tri-hydrogen is formed from the $H_\beta H_\beta H_O$ composition, then $H_\alpha H_\alpha$ is strongly favored over $H_\beta H_\alpha$ double hydrogen elimination. The dominant $H_\alpha H_\alpha$ double hydrogen elimination accompanying the specific $H_\beta H_\beta H_O$ tri-hydrogen formation in the $H_3^+ + C_2HO^+ + 2H$ channel is a particularly complex process, which might be initiated by the escape of a roaming $H_2$ from the $\alpha$ site, but the resulting tri-hydrogen also requires subsequent bond rearrangement.

The horizontal spread of the kernel density functions in the bottom set of violin plots of Fig. 4d illustrates that the fit is becoming slightly unstable, with some combinations producing zero probability for one 2H combination, driving the other linked 2H probability to one. Even with this instability, useful information can be recovered. The violin plots for all the incomplete $H_3^+$ breakup channels, including the initial sites of the ejected hydrogen atoms, are provided in SN 9.

### Opportunities for theoretical comparisons

The probability that an initial hydrogen site of an ethanol dication would contribute to the formation of a particular hydrogen-rich fragment was measured by combining the results from coincidence ion momentum imaging measurements of seven deuterium-tagged isotopologues of ethanol. These quantitative results provide insights into previously unresolved questions surrounding hydrogen migration in ethanol. For example, $H_\alpha H_\alpha H_O^+$ formation is far more likely than $H_\alpha H_\alpha H_\beta^+$, indicating that the roaming $H_\alpha H_\alpha^2$ tends to sample the hydroxyl side of the molecule more than the methyl side. Hydrogen scrambling occurs in several channels, including $H_2O^+$ formation, and hydronium ions predominantly form through migration of independent hydrogen atoms to the hydroxyl group, as suggested by Kling and co-workers[23]. Furthermore, the significant differences observed in the measured site-specific probabilities associated with complete and incomplete breakup channels calls for caution when generalizing measurements limited to complete breakup channels to the overall picture of the molecular fragmentation. The key to reaching these conclusions is the quantitative determination of the probability associated with hydrogen migration from each site.

By eliminating experimental ambiguities about the initial site of the hydrogen atoms participating in the hydrogen migration process, the data presented above put stringent constraints on theoretical modeling of intramolecular hydrogen dynamics. Molecular dynamics calculations of the fragmentation of the ethanol dication can now be benchmarked against experimentally determined probabilities for initial-site compositions yielding the hydrogen-rich molecular fragments. Additional composition-specific characteristics of the molecular fragments, such as KER or angular distributions, can also be extracted from the data and provide more detailed tests for future calculations of the underlying dynamics. The method can be extended to more complex molecular ions, and the anticipated collaborative efforts between experiment and theory should enhance the fundamental understanding of the coupling between electronic and nuclear dynamics in polyatomic molecular ions which, in turn, can be applied to neutral molecules.

## Methods

### Experimental technique

Laser pulses of up to 2 mJ per pulse at a repetition rate of 10 kHz were provided by a Ti:sapphire laser system, known as PULSAR[65]. The pulses had a 23 fs duration (FWHM in intensity) and a central wavelength of 790 nm. The attenuated laser pulses were focused by a $f = 75$ mm spherical mirror to a calibrated peak intensity of $3.0 \times 10^{14}$ W cm$^{-2}$. The peak intensity was selected to maximize the amount of double ionization while at the same time keeping the triple-ionization rate negligible. Under these conditions, the production of ethanol dimers is negligible, as shown in SN 12.

The laser pulse parameters were monitored in a variety of ways, as detailed below, to ensure that conditions remained the same throughout the measurements of all ethanol isotopologues. First, the laser peak intensity was monitored on a shot-to-shot basis via the amplitude of a photodiode signal. The time-averaged power was also monitored separately using a power meter. Both the power meter and the photodiode signals were recorded as part of the event-mode data acquisition using an analog-to-digital converter (ADC). The ADC data was used to exclude events obtained from unsatisfactory laser pulses in analysis. The laser power was controlled with a combination of a polarizing beam splitter cube and a waveplate, and slow drifts in the laser power were corrected during the experiment by pausing data collection and resetting the power. Second, the beam pointing into the ultra-high vacuum (UHV) chamber was monitored in angle and position using two cameras on an equivalent path traversing the optics table. The peak laser intensity was determined by measuring the recoil momentum distribution of Ne$^+$ ions along the laser polarization and locating the point associated with the $2U_p$ kinetic energy of the electron, where $U_p$ is the pondermotive energy[66]. Finally, the pulse duration and bandwidth of the pulses were monitored throughout the measurements using second-harmonic-generation frequency-resolved-optical-gating (SHG-FROG)[67].

The incident laser beam polarization was set parallel to the spectrometer's time-of-flight (TOF) axis of our COLTRIMS setup (details of this apparatus are described in refs. [68,69]). The three-dimensional momenta of the ions produced from ethanol were evaluated from their detected time and position using COLTRIMS methodology[49–51], while the electrons were not detected. The CTOF spectra (shown, e.g., for $CD_3CH_2OH$ in Figs. 2 and 3 of SN 6) were used to identify the two-body breakup channels (both complete and incomplete)[2].

Ethanol vapor from commercially-obtained high-purity liquid samples (98–99.5% D, as detailed in SN 1) was introduced to the UHV chamber, where it expanded into the vacuum as a supersonic gas jet that was collimated by skimmers prior to intersection with the laser focus. To clean the gas lines of the previous sample, we ran neon into the gas manifold at approximately 1 atmosphere while baking the gas lines for 1–2 h. The laser beam remained on the neon jet target during baking. The resulting ions were recorded and used to assess the level of contamination in the manifold. Once the previous isotopologue of ethanol was not visible in the time-of-flight spectra, the gas lines were allowed to cool, and a different isotopologue of ethanol was introduced into the system from a new bubbler connected to the manifold.

### Data analysis

The data analysis consists of evaluating the branching ratios of the ethanol dication fragmentation channels, and then using them to determine the relative site-specific probabilities of the hydrogens making up the final product.

### Branching ratios

The initial steps of the COLTRIMS data analysis are the same as we have described in earlier publications[56] and detailed further in SN 6. In short, dissociative ionization channels are identified on a CTOF map. The spectrometer parameters, laser-molecule interaction point, and target gas jet velocity are calibrated by considering all coincidence channels simultaneously and by using the expected symmetries about the laser polarization. For complete two-body breakup channels, momentum conservation is used to separate the desired events from background.

In order to isolate the true coincidence events of interest, i.e., ion-pairs originating from the same ethanol molecule in our case, one must subtract all false coincidence events. The false coincidence events that arise from ionization of two different parent molecules by the same pulse were mimicked by randomly pairing individual ion counts from different laser shots, thus generating a sample of purely false-coincidence events. The reproduced distribution of false coincidences is then scaled to match a channel that can arise only from a false coincidence. Finally, we subtracted the scaled artificially-generated false coincidences from the measured spectrum yielding the true coincidence spectrum[70].

In the case of complete two-body breakup channels, the number of events in each channel is converted to a branching ratio as shown in Eq. (1), for example, for the $D_3^+ + C_2H_3O^+$ breakup channel of $CD_3CH_2OH$,

$$R_3\left(D_3^+\right) = \frac{N\left(D_3^+ + C_2H_3O^+\right)}{\sum\limits_{all} N_C(m_1, m_2) + \sum\limits_{all;\, m_n \leq 6} N_{\bar{C}}(m_1, m_2; m_n)}. \tag{1}$$

In Eq. (1), $R_i(m_1)$ is the branching ratio for the $m_1$ breakup channel of the $i$th ethanol isotopologue as enumerated in Fig. 1. $N_C(m_1, m_2)$ is the number of measured ion-pairs from the complete channel with $m_1$ and $m_2$ being the mass of the first and second ions, respectively. Similarly, $N_{\bar{C}}(m_1, m_2; m_n)$, is the number of measured ion-pairs associated with an incomplete breakup channel with undetected neutral fragment(s) having a mass $m_n$. Since we ultimately compare probabilities across complete- and incomplete-breakup channels, we normalize each

channel to the sum of both measured complete- and incomplete-breakup channels, given by the first and second terms in the denominator of Eq. (1), respectively. Note that the second sum is truncated, i.e., $m_n \leq 6$, as the loss of more massive fragments hinders the separation of these breakup channels from other channels. It is worth mentioning that in studies focused solely on complete breakup channels (e.g., refs. [2,23]), the sum of incomplete channels in Eq. (1) is dropped, simplifying the analysis and yielding branching ratios with smaller relative errors.

### Site-specific probabilities

Neglecting isotopic effects, the branching ratio is the sum of the relevant site-specific probabilities for each isotopologue. For example, the $D_3^+ + C_2H_3O^+$ breakup channel of $CD_3CH_2OH$ yields

$$R_3(D_3^+) = P_{\beta\beta\beta}, \tag{2}$$

where $P_{\beta\beta\beta}$ is the site-specific probability of forming a tri-hydrogen ion from the $H_\beta H_\beta H_\beta$ initial-site composition. Unlike Eq. (2), most measured branching ratios include contributions from more than one site, such as

$$R_6\left(D_3^+\right) = P_{\beta\beta\beta} + P_{\beta\beta\alpha} + P_{\alpha\alpha\beta} \tag{3}$$

and

$$R_6\left(HD_2^+\right) = P_{\beta\beta O} + P_{\beta\alpha O} + P_{\alpha\alpha O}. \tag{4}$$

for the $CD_3CD_2OH$ isotopologue. Note that the site-specific probabilities above are defined to include the multiplicity of the different initial-site combinations, e.g., there is only one way to combine $H_\beta H_\beta H_\beta$ but there are six ways to combine $H_\beta H_\alpha H_O$.

Some measured channels cannot be associated with a unique ion pair because of mass overlaps, like when $H_3^+$ and $HD^+$ can both be produced from the same ethanol isotopologue (see SN 2 for complete list). The branching ratios of such channels are not used in the analysis that follows. By avoiding channels with mass overlaps, we avoid having to make any assumptions that different hydrogen sites behave in the same way, as are sometimes invoked in other cases[41,42]. For tri-hydrogen ion formation, however, the seven measured isotopologues (No. 1-7) yield 10 "clean" equations of the type illustrated by Eqs. (2)–(4) (the complete set of equations is listed in SN 3). These 10 equations form an overdetermined set because there are only six possible initial-site compositions for tri-hydrogen ion formation: $P_{\beta\beta\beta}$, $P_{\alpha\alpha O}$, $P_{\beta\alpha O}$, $P_{\beta\alpha\alpha}$, $P_{\beta\beta O}$, and $P_{\beta\beta\alpha}$. We label the sum of these probabilities, which is the probability for $H_3^+$ production, $P(H_3^+)$ (see Fig. 3).

The six site-specific probabilities for tri-hydrogen ion formation are determined by optimizing a least-squares fit of these probabilities to the ten experimentally determined branching ratios (see results in Fig. 3a and SN 7). To evaluate the uncertainty due to this fitting procedure, the fit is repeated, in a Monte Carlo fashion, by sampling branching ratios from a normal distribution centered around the measured values and with a width determined from the experimental uncertainties. The characteristics of the normal distribution representing the measured branching ratios are, in turn, determined from a combination of the statistical and systematic uncertainties in the branching ratio measurement (as detailed in SN 8). The Monte Carlo uncertainty propagation typically contained 10,000 samples. Larger sample sizes did not change any results significantly.

The analysis for complete two-body breakup channels leading to hydronium, water, and methane ions is similar except that mass overlaps couple the branching ratio equations. For example, ions with $m/q = 20$ from $CD_3CH_2OH$ (isotopologue #3) include contributions from water and hydronium ions, specifically $D_\beta H_\alpha H_\alpha O$, $D_\beta H_\alpha H_O O$, and $D_\beta D_\beta O$, therefore, the site-specific probabilities contributing to this

branching ratio can be written as

$$R_3(m_1/q_1 = 20) = P_{\beta\beta} + P_{\beta\alpha\alpha} + P_{\beta\alpha O}. \qquad (5)$$

Since the equations for hydronium, water, and methane ions are coupled in the manner of Eq. (5), we need to solve for all three species simultaneously. The resulting system of equations, however, is still overdetermined, with the seven isotopologues producing 22 branching ratios that define 16 site-specific probabilities (see SN 4). Fitting an expanded system of equations including $CH_5^+$ had a negligible effect on all final site-specific probabilities because the $CH_5^+ + CHO^+$ yield was only $\approx 11\%$ of the $CH_4^+ + CH_2O^+$ yield. Therefore, the $CH_5^+$ channels were not included in the set of equations leading to the reported results.

For incomplete coincidence channels, the general analysis method is unchanged. The important difference, however, is that there are additional site-specific compositions to determine, since the sites of the ejected hydrogen atoms are now of interest. There are, in turn, additional branching ratios that can be measured to determine the initial-site compositions. Due to mass overlaps, many of the branching ratios contribute to multiple probabilities. We found that the best least-squared fits were associated with using the largest collection of measured branching ratios (i.e., a set of 54 equations listed in SN 5) to define the 36 possible site-specific probabilities arising from the combination of the $H_3^+ + C_2H_2O^+ + H$ and $H_3^+ + C_2HO^+ + 2H$ channels. The identity of the undetected neutral fragment is recovered from mass balance, and we cannot distinguish, for example, elimination of a $H_2$ molecule from elimination of two hydrogen atoms. Although not explicitly used in this work, the momentum of the dissociating neutral atoms can be reconstructed from momentum conservation. Additional dissociation combinations, involving heavier neutral fragments, could be included in this analysis if needed. At the present time, we limited the analysis to the region of the CTOF map where the identifications of the incomplete coincidence channels are more manageable.

### Reporting summary
Further information on research design is available in the Nature Portfolio Reporting Summary linked to this article.

## Data availability
The data that support the findings of this study are available in this article and its Supplementary Information. The raw, or suitably reconstructed, data are available from the corresponding authors upon request. See Supplementary Note 10 for a description of the files structure. Source data are provided with this paper.

## Code availability
The least-squares fitting and Monte Carlo uncertainty propagation code are available in the Supplementary Software package.

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

## Acknowledgements

We thank C.W. Fehrenbach for assistance with the PULSAR laser. The U.S. Department of Energy (DOE), Office of Science, Office of Basic Energy Sciences, Chemical Sciences, Geosciences, and Biosciences Division under Award No. DE-FG02-86ER13491 supported the J.R. Macdonald Laboratory (JRML) and T.S., B.K., P.F., B.J., K.B., F.Z., K.R.P., K.D.C., D.R., A.R., and I.B.-I. The PULSAR laser at JRML was provided by grant DE-FG02-09ER16115 from the same funding agency. National Science Foundation Grant No. PHY-2309192 supported E. Weckwerth and E. Wells. E. Weckwerth received additional funding from the Augustana

University Viste Student/Faculty Research Fellowship and the South Dakota Space Grant Consortium. The U.S. DOE, Office of Science, Office of Basic Energy Sciences, Chemical Sciences, Geosciences, and Biosciences Division under Grant No. SISGR (DE-SCO002325) supported M.D.

## Author contributions

T.S. planned the experiment with input from M.D., D.R., A.R., and I. B.-I. The experiment was setup and performed by T.S., B.K., P.F., B.J., F.Z., K.B., K.R.P., K.D.C., D.R., and A.R.; E. Weckwerth analyzed the data with guidance and support from T.S. and evaluated the site-specific probabilities guided by E. Wells and I.B.-I.; E. Wells, K.D.C., D.R., A.R., and I.B.-I. mentored the students and post-docs; E. Weckwerth, E. Wells, and I.B.-I. prepared the manuscript, which all co-authors approved.

## Competing interests

The authors declare no competing interests.
