## [Peer Review File · Nature Communications]

Initial-site characterization of hydrogen migration following strong-field double-ionization of ethanolREVIEWER COMMENTS

Reviewer #1 (Remarks to the Author):

The manuscript "Initial-site characterization of hydrogen migration in ethanol" by Severt et al. discusses the mechanism by which different products are formed following the double ionization of ethanol. Overall, the manuscript is well written. The results clearly presented, previous work is appropriately cited, and the implications of the findings are discussed. I have several comments for the authors, set out below. A few of these are very minor. While the first point is quite general in nature, it is perhaps the most important to address as it relates to how generally the results can be applied.

1. All measurements are undertaken on the properties of the ethanol dication, and not the neutral species as implied by the title. The introduction begins by noting that the migration of H within polyatomic molecules is a common process in a variety of fields, without addressing why this is of specific interest in the ethanol dication (or any other dicationic species). Later in the introduction, reference is made to previous work on the ethanol dication, so perhaps that is the motivating factor for this research? I'm not convinced that there is a meaningful link between the very general statements about the ubiquity of H migration "within polyatomic molecules" at the start of the introduction, and the specific interest in the ethanol dication (or even dications more generally) that follows. No evidence is provided to support the implication that neutral ethanol would exhibit the same behaviour as seen in the dication. As such, the title appears to be rather misleading, with the start of the introduction largely irrelevant.

2. Another general comment relates to the statement "...establishing that the roaming H_αH_α moiety is 15 times more likely to abstract the hydroxyl proton than a methyl-group proton..." I took this to mean that the authors independently established the presence (and relative importance) of one or more roaming pathways. As far as I understand the paper, this is not the case. In the manuscript, earlier studies are cited as having identified possible roaming pathways in this system. However, the measurements undertaken in this work cannot establish the presence or absence of a roaming mechanism versus a more traditional abstraction pathway. The comment in the abstract should be removed or clarified.

3. On page 2, in the third results paragraph, it is stated "the fundamental assumption is that the mass difference between H and D does not significantly affect the properties to be studied...verified in our case by comparing the branching ratios from all-hydrogen..." Data comparing the fully hydrogenated and fully deuterated isotopologues are provided, with figure 2 supposedly showing "the relative deviation from the mean was less than 10%." I do not understand what the authors are referring to here. What mean? What relative deviation? The statement is repeated in the figure caption, without explanation of what mean value is being referred to, or why having a relative deviation from the mean below 10% is important.

4. In figure 2, the error bars are not defined in the caption. The error bars also appear to sit entirely on top of the bar plots. Was this intentional?

5. I agree with the authors that the distribution of products into these four channels appears to be quite similar for both fully deuterated and fully hydrogenated. As comparing the branching between these four channels for the fully hydrogenated and fully deuterated isotopologues is a key assumption underpinning all subsequent analysis, it would be helpful if the authors conducted some further analysis on this data. Is there a reason the results couldn't be re-normalised (with respect to the sum of these four product channels of interest) to facilitate a more in-depth comparison?

6. The authors mention that the all-D and all-H branching ratios "are quite similar in magnitude and resemble the size of mass effects previously observed from other small molecules." Have any potential energy surface calculations been conducted, to look at the relative energies of the different product channels for the various isotopologues? There are several assumptions being

invoked here, justified based on observations in other systems. While the assumptions may well be valid, the conclusions would have more weight if supported by electronic structure calculations and detailed analyses of how the potential energy landscape changes with deuteration at the various H sites.

7. There is a detailed discussion of the various H sites in the ethanol dication and an in-depth analysis of what contribution of each H site has to a given product. At several places in the manuscript (in the first instance, on page 4 in the caption to figure 3), reference is made to "Monte Carlo output" when discussing the uncertainty in the results. However, the only description I can find of the Monte Carlo simulations is a single sentence on page 9, where it is stated that "The Monte Carlo simulations typically contained 10,000 samples." Very few additional details are provided in the supplementary information, with the relevant paragraph on page 14 opening with the statement "As described in the main article..." This point requires further clarification in the main body of the manuscript, as it is a critical part of the analysis.

8. Note that I am not an expert in the experimental methodology adopted in this work. I am not qualified to comment on the veracity of the experimental approach, and I will leave this to other reviewers. As a general comment, it would be useful for the authors to briefly note in the main manuscript what the strengths and limitations of the method are.

9. It is stated in the methods section that "high-purity liquid samples" of ethanol were used. The purity of the various isotopologues should be stated. When swapping the isotopologues of ethanol, how were impurities (introduced by breaking vacuum) removed from the gas lines? How was the laser used to assess the level of contamination in the manifold?

10. I was missing a general conclusion to the findings. What are the key take-home messages that the authors want to convey? Given that I found parts of the abstract and introduction to be misleading, I was looking for a summary of what the paper was actually about. A clear conclusion appears to be missing.

Reviewer #2 (Remarks to the Author):

The manuscript reports on experiments investigating the strong-field induced dissociation of ethanol. An experimental procedure is described for determining the relative probabilities of the formation of fragment ions containing hydrogen, where the hydrogen atoms start at different positions in ethanol. The key idea is to collect data for all types of deuterated molecules, and to determine the probabilities of the different pathways by an elaborate fitting procedure.

I think the manuscript is a very interesting one which would be well suited for publication in Nature Communications. The method should have a large impact on the field of strong-field chemistry and molecular physics.

Provided that the authors can reply to my comments below, I would recommend the manuscript for publication.

1. In Fig. 2, the branching ratios of ethanol and completely deuterated ethanol are compared. The difference is not large, so we can rule out quantum tunneling as the dissociation mechanism. Would it be possible to discuss quantitatively the difference in the branching ratios by referring to the mass difference? The difference could be a factor of $1/\sqrt{2}$?

2. In the introduction and in the discussion of H₃⁺ formation on page 5. I think that the paper [J. Chem. Phys. 139, 181103 (2013) <https://doi.org/10.1063/1.4830397>] should be mentioned in the discussion.

3. I would appreciate if the "violin plots" could be explained briefly. Even if this type of plots is

explained in [50], it would be helpful for the reader if a brief explanation could be provided to make the manuscript self-contained.

4. As I understand the manuscript, the experimental method and analysis is based on the comparison of data obtained by different isotopologues, and the assumption that the branching ratios for the different channels are the same for different isotopologues. However, the data in Fig. 2 shows that the branching ratios are not the same. Is the difference of the branching ratios somehow taken into account in the estimation of the error bars, or the shape of the violin plots?

Minor remark:

1. In the abstract, an "H_alpha H_alpha" is mentioned, but this notation is defined later on in the paper, and cannot be understood only by reading the abstract. A description which can be understood on its own would be preferred.

Reviewer #3 (Remarks to the Author):

In this manuscript by Severt et al., the authors determined experimentally the originating sites of hydrogen in the formation process of tri-hydrogen cation from strong field induced ethanol dications. In this study, the authors used seven isotopologues of ethanol with deuterium and hydrogen located at different sites. With PIPICO measurements carried out by a COLTRIMS apparatus, branching ratios of many dissociation pathways were identified and a fitting process was adopted to extract the hydrogen initial site information. This is an interesting work and add to a large body of existing knowledge on dissociative double ionization of various small molecules. The employed method is not new because isotopologues of ethane have been previously used to extract similar information (see ref. 45 and also in JCP 134, 064324 (2011)). The current work deals with a larger molecule and more complicated dissociation pathways. But the results seem similar, i. e. that hydrogen scrambling is a significant process in strong field induced dissociation processed. The work is surely publishable, but I have no strong opinions regarding whether it is suitable for Nature Communications. I see it can go both ways. But I do have some comments for the authors to consider during a revision, if offered.

1. This is an experimental work. But the first sentence of the abstract gives the impression that it might include theory work, which is not true. I think the author should rewrite or remove that sentence.
2. The C-C bond breaking channel was not discussed. This is a main channel and will reveal the degree of hydrogen scrambling. The authors should include it in Fig. 2 also.
3. There is serious congestion in the PIPICO spectrum. The authors explained they used momentum conservation to select intended channels. How robust are the branching ratios when the momentum conservation criterion is varied (sum $P < ?$).
4. The authors should cite JCP 134, 064324 (2011).

We thank all the reviewers for constructive feedback that we believe has led to a stronger revised manuscript. The beginning and the end of the article, in particular, have been rewritten to more explicitly address what we see as the main attributes of this work and the usefulness to the larger community. In addition, a new section in the Supplemental Information (Sec. 13) addresses isotopic differences in detail. We hope that these efforts will be well-received by the reviewers and we look forward to the article being accepted.

Over the following pages, we will respond to each point raised by the three reviewers. The color scheme used in our response is

- black text indicates the reviewer comments
- blue text is our response to the reviewer comments
- green text represents quotes from the updated manuscript or the Supporting Information.

REVIEWER COMMENTS

Reviewer #1 (Remarks to the Author):

The manuscript “Initial-site characterization of hydrogen migration in ethanol” by Severt et al. discusses the mechanism by which different products are formed following the double ionization of ethanol. Overall, the manuscript is well written. The results clearly presented, previous work is appropriately cited, and the implications of the findings are discussed. I have several comments for the authors, set out below. A few of these are very minor. While the first point is quite general in nature, it is perhaps the most important to address as it relates to how generally the results can be applied.

We thank Reviewer #1 for the positive comments and respond to each individual point below.

1. All measurements are undertaken on the properties of the ethanol dication, and not the neutral species as implied by the title. The introduction begins by noting that the migration of H within polyatomic molecules is a common process in a variety of fields, without addressing why this is of specific interest in the ethanol dication (or any other dicationic species). Later in the introduction, reference is made to previous work on the ethanol dication, so perhaps that is the motivating factor for this research? I’m not convinced that there is a meaningful link between the very general statements about the ubiquity of H migration “within polyatomic molecules” at the start of the introduction, and the specific interest in the ethanol dication (or even dications more generally) that follows. No evidence is provided to support the implication that neutral ethanol would exhibit the same behaviour as seen in the dication. As such, the title appears to be rather misleading, with the start of the introduction largely irrelevant.

We agree with the reviewer that this work is done on the ethanol dication while many of the applications of hydrogen migration involve neutral molecules. To avoid possible misleading interpretation of the title, pointed out by the reviewer, we have modified the title to “Initial-site characterization of hydrogen migration following strong-field double-ionization of ethanol”.

Furthermore, in the abstract we explicitly state that our measurements occur following double ionization.

We demonstrate that coincidence ion-imaging measurements of a few deuterium-tagged isotopologues of ethanol can determine the contribution of each initial-site composition to hydrogen-rich fragments following strong-field double ionization.

As for the opening paragraph introducing the reader to the wide range of phenomena involving hydrogen migration, we think it has a value. Given the reviewer comments, however, it seems that we failed to convey how our experimental work may impact the understanding of hydrogen migration in a broad sense, i.e., beyond the specific ethanol dication. The key is that theoretical approaches to molecular dynamics are used to describe hydrogen migration processes in both neutral and charged molecules. Therefore, providing theorists with challenging benchmark data, like the site-specific probabilities we have measured, may lead to advancements of these theoretical approaches, and in turn those will have impact on understanding hydrogen migration in a broad range of applications involving mainly neutral molecules. We have modified the introduction significantly, and some of the changes address this point, for example by saying: “Comparisons between theoretical calculations and measurements of ionic molecular fragments are important because they benchmark theoretical approaches that are also applied to the neutral molecules in which many applications occur.” In addition, we highlight this point in the last paragraph before the “Methods” section, which starts with “By eliminating experimental ambiguities about the initial site of the hydrogen atoms participating in the hydrogen migration process, ...”.

We return to this theme near the end of the Results section, concluding with

The method can be extended to more complex molecular ions, and the anticipated collaborative efforts between experiment and theory should enhance the fundamental understanding of the coupling between electronic and nuclear dynamics in polyatomic molecular ions and, in turn, be applied to neutral molecules.

Finally, we note that the goal of our study, described above, is easier to attain using fragmentation of molecular dications, although it is still challenging. This point is now introduced explicitly in the first paragraph.

“Despite the fact that many of these applications appear predominantly in neutral molecules, most recent gas-phase experiments use ions to probe the dynamics of hydrogen migration mechanisms, including roaming [1, 2, 9, 10, 19–27]. Ion-based experiments are favored in these situations because they generally have more available observables.”

We also agree with the reviewer that ethanol dications have attracted considerable attention in their own right, as illustrated by two recent publications in Nature Communications (Refs. [2,23]). This was one of the reasons we picked ethanol as the three non-equivalent hydrogen sites molecular target for our study, and used our results to explore also if our measured site-specific probabilities are consistent with hydrogen migration mechanisms proposed in previous publications. The latter point is discussed, for example, in the paragraph starting with “These comprehensive site-specific results yield significant additional insights into hydrogen migration and bond rearrangement in the dissociation of ethanol dications, resolving questions from previous work...” (page 2), as well as the 1st paragraph in “Opportunities for theoretical comparisons” starting with “The probability that an initial hydrogen site of an ethanol dication would contribute to the formation of a particular hydrogen-rich fragment...” (page 8).

2. Another general comment relates to the statement "...establishing that the roaming H_αH_α moiety is 15 times more likely to abstract the hydroxyl proton than a methyl-group proton..." I took this to mean that the authors independently established the presence (and relative importance) of one or more roaming pathways. As far as I understand the paper, this is not the case. In the manuscript, earlier studies are cited as having identified possible roaming pathways in this system. However, the measurements undertaken in this work cannot establish the presence or absence of a roaming mechanism versus a more traditional abstraction pathway. The comment in the abstract should be removed or clarified.

We did not intend to infer this conclusion, but we thank the reviewer for pointing out that it could indeed be interpreted this way. We have rewritten the abstract to clarify this. The modified sentence reads

"...establishing that the central two hydrogen atoms are 15 times more likely to abstract the hydroxyl proton than a methyl-group proton..."

This revision also removes the introduction of the H_αH_α notation in the abstract, which was commented on by Reviewer #2.

3. On page 2, in the third results paragraph, it is stated "the fundamental assumption is that the mass difference between H and D does not significantly affect the properties to be studied...verified in our case by comparing the branching ratios from all-hydrogen..." Data comparing the fully hydrogenated and fully deuterated isotopologues are provided, with figure 2 supposedly showing "the relative deviation from the mean was less than 10%." I do not understand what the authors are referring to here. What mean? What relative deviation? The statement is repeated in the figure caption, without explanation of what mean value is being referred to, or why having a relative deviation from the mean below 10% is important.

We removed the reference to the "mean" in the main article and provided a more comprehensive definition in Section 13 of the Supplemental Information (SI), which we reference in the text. The new SI Section 13 is discussed more fully in item #6 below.

4. In figure 2, the error bars are not defined in the caption. The error bars also appear to sit entirely on top of the bar plots. Was this intentional?

The error bars are symmetric and represent the combined statistical and systematic uncertainties, as described by the modified caption.

...The error bars are the combination of the systematic and statistical uncertainties as described in Section 8 of the Supplemental Information (SI). ...

The figure has been replotted so that the error bars fall over the color bars.

5. I agree with the authors that the distribution of products into these four channels appears to be quite similar for both fully deuterated and fully hydrogenated. As comparing the branching between these four channels for the fully hydrogenated and fully deuterated isotopologues is a key assumption underpinning all subsequent analysis, it would be helpful if the authors conducted some further analysis on this data. Is there a reason the results couldn't be re-

normalised (with respect to the sum of these four product channels of interest) to facilitate a more in-depth comparison?

As mentioned above, we have added a new section to the SI to discuss this topic in detail. The reviewer's suggested method of renormalization is included as part of that analysis and shown in Fig. S10 in SI Section 13.A. An additional response to this point is included after item #6 below.

6. The authors mention that the all-D and all-H branching ratios “are quite similar in magnitude and resemble the size of mass effects previously observed from other small molecules.” Have any potential energy surface calculations been conducted, to look at the relative energies of the different product channels for the various isotopologues? There are several assumptions being invoked here, justified based on observations in other systems. While the assumptions may well be valid, the conclusions would have more weight if supported by electronic structure calculations and detailed analyses of how the potential energy landscape changes with deuteration at the various H sites.

We agree with the notion that having a theoretical counterpart, both structure and dynamics calculations, will strengthen the conclusions – in a way, this is really the motivation driving our work as it seems that significant improvements, especially of molecular dynamics theory, are needed, and providing quantitative benchmark results may aid accomplish that. This also holds for isotopic effects, which are usually harder to track. That said, however, our experience shows that the energy differences are usually small and dynamic isotopic phenomena are usually governed by changes in Franck-Condon factors due to differences in the nuclear wave functions and differences in the speed between H and D.

It is also important to note that isotopic effects were not the focus of our work; in a way, the fact that they are usually small is key to deuterium tagging. We carefully analyzed our data to verify that isotopic differences will not affect the site-specific probabilities, which are the goal of our work. We significantly expanded the discussion of this issue by adding an entirely new section to the SI (Sec. 13), which provides the reader with ample evidence that the isotopic differences in the measured branching ratios are at the level of estimated error of the evaluated site-specific probabilities. For example, the deviation from the mean (or significance, S , as defined in Eq. S10) between the two-body branching ratios from the all-H (#1) and all-D (#7) isotopologues of ethanol is about 1σ , where σ is the combined statistical and systematic uncertainty, as shown in Fig. S9 (see SI, Sec. 13). Furthermore, where possible, we looked at other isotopologues to see if the isotopic differences there may cause larger deviations (SI Sec. 13.B), and verified that overall, the trend is that the small branching ratios variations are not modifying the resulting site-specific probabilities beyond their estimated errors (SI Sec. 13.C).

7. There is a detailed discussion of the various H sites in the ethanol cation and an in-depth analysis of what contribution of each H site has to a given product. At several places in the manuscript (in the first instance, on page 4 in the caption to figure 3), reference is made to “Monte Carlo output” when discussing the uncertainty in the results. However, the only description I can find of the Monte Carlo simulations is a single sentence on page 9, where it is stated that “The Monte Carlo simulations typically contained 10,000 samples.” Very few additional details are provided in the supplementary information, with the relevant paragraph on page 14 opening with the statement “As described in the main article...” This point requires further clarification in the main body of the manuscript, as it is a critical part of the analysis.

In this case, the original choice of the word “simulation” may have implied that there was more involved in this method than we intended. The technique is a simple Monte Carlo error propagation, which is described in, for example, Chapter 5 of the textbook by Bevington and Robinson (2003). We added a more explicit explanation to the SI in Section 8, along with some supporting references about the general idea of Monte Carlo error propagation. In short, however, a distribution of fitting inputs (the branching ratios) is generated. The widths of these input distributions are equivalent to the experimental uncertainty in the branching ratio, as shown in SI Fig. S4. Then a random value from each input is selected and used to fit the system of equations. The process is repeated, in a Monte Carlo fashion, and the results are tabulated, yielding the output distributions illustrated by the violin plots.

We have replaced the phrase “Monte Carlo simulation” with “Monte Carlo uncertainty propagation” in the text, which should give a more appropriate description of what is being done. The relevant portion of the Methods Section now reads.

To evaluate the uncertainty due to this fitting procedure, the fit is repeated, in a Monte Carlo fashion, by sampling branching ratios from a normal distribution centered around the measured values and with a width determined from the experimental uncertainties. The characteristics of the normal distribution representing the measured branching ratios are, in turn, determined from a combination of the statistical and systematic uncertainties in the branching ratio measurement (as detailed in Sec. 8 of the SI). The Monte Carlo uncertainty propagation typically contained 10,000 samples. Larger sample sizes did not change any results significantly.

8. Note that I am not an expert in the experimental methodology adopted in this work. I am not qualified to comment on the veracity of the experimental approach, and I will leave this to other reviewers. As a general comment, it would be useful for the authors to briefly note in the main manuscript what the strengths and limitations of the method are.

The major advantage of this experimental approach is that it extends deuterium tagging to specify the contributions from three or more non-equivalent hydrogen sites in a molecule. This has not been done (to our knowledge) previously because of the inherent problem of identifying more than two sites with only H and D. We now outline this explicitly in the introduction.

Expanding this work beyond molecules with two nonequivalent hydrogen sites is a challenge since there are only two readily available isotopes of hydrogen, and thus no single deuterium-tagged experiment can identify the role of all hydrogen sites.

Shortly thereafter, we summarize how this problem is overcome.

In this article, we demonstrate that a combination of ion coincidence momentum-imaging (COLTRIMS) [49–51] measurements, using a few deuterium-tagged isotopologues of ethanol, can determine from which molecular sites the hydrogen atoms that compose a molecular fragment originated.

In the second paragraph of the right column on page 2, we inserted a statement highlighting the need for momentum imaging to obtain the incomplete channels.

The ability to reconstruct the net momentum of the neutral fragments is one of the major experimental advantages of the coincidence ion momentum imaging technique.

Coincidence-time-of-flight could be used with similar effect, but the full momentum imaging is required to obtain site-specific values of dynamical variables, such as KER, and it helps with the identification of the incomplete channels. As far as we are aware, there is no other experimental technique that can achieve similar results. This is noted in the first full paragraph on page 4.

Since the fragment momentum is calculated on an event-by-event basis, associated dynamic quantities can be calculated and binned. Provided there are sufficient statistics in each bin, the relative site-specific probability of a dynamic quantity can be calculated for each bin in the same manner as the relative probability of the composition of the entire fragmentation channel.

The drawback is the duration and technical requirements of the experiment and the analysis. It takes a long time to build up the statistics needed for all the different isotopologues, and it takes an even longer time to conduct a careful analysis of all the dissociation channels needed to obtain the branching ratios. The first paragraph on page 4 now points out that the duration of the experiments is the major drawback to employing the technique for larger molecules.

Hydrogen migration and bond-rearrangement processes in these larger molecules have recently garnered attention [1, 2, 60, 61]. The main experimental impediment to exploring these larger systems is the time needed to obtain data for all isotopologues, but laser systems running at hundreds of kilohertz [62] make these potential experiments more feasible.

9. It is stated in the methods section that “high-purity liquid samples” of ethanol were used. The purity of the various isotopologues should be stated. When swapping the isotopologues of ethanol, how were impurities (introduced by breaking vacuum) removed from the gas lines? How was the laser used to assess the level of contamination in the manifold?

The purities and commercial sources of the deuterated compounds are now listed in Table S1 of the SI. The basic information is stated in the main manuscript in the Methods section, left column of page 9.

Ethanol vapor from commercially-obtained high-purity liquid samples (98—99.5% D, as detailed in Sec. 1 of the SI) was introduced to the UHV chamber,...

We have clarified how we verified the contamination due to the previous ethanol isotopologue is negligible by adding the following text in the paragraph starting with “Ethanol vapor from high-purity liquid samples...”, in the Methods section, specifically saying: “To clean the gas lines of the previous sample, we ran neon into the gas manifold at approximately 1 atmosphere while baking the gas lines for 1-2 hours. The laser beam remained on the neon jet target during baking. The resulting ions were recorded and used to assess the level of contamination in the manifold. Once the previous isotopologue of ethanol was not visible in the time-of-flight spectra, the gas lines were allowed to cool, and a different isotopologue of ethanol was introduced into the system from a new bubbler connected to the manifold.”

Finally, it is worth mentioning that small isotopic impurities are not a major concern in this experiment for the following reasons. First, in the complete fragmentation channels, all the ions are measured, so the identity of the parent ion is known. Second, in cases where there is some doubt if a ^1H and ^2D are swapped, then we have avoided using that channel and used only the channels we had confidence about, i.e., we have used only “clean” channels and still had an overdetermined set of equations to solve (fit).

10. I was missing a general conclusion to the findings. What are the key take-home messages that the authors want to convey? Given that I found parts of the abstract and introduction to be misleading, I was looking for a summary of what the paper was actually about. A clear conclusion appears to be missing.

As the reviewer is probably aware, the Nature Communications formatting guidelines do not allow for a specific summary or conclusion. We tried to use the paragraph following the “Opportunities for theoretical comparisons” heading to serve this purpose in the original draft. We have now rewritten it to echo many of the main points at the end of the introduction, before pointing out the resulting challenge for theoreticians. Together, these form the take-home message: this general method removes the experimental ambiguity often associated with the initial site of a hydrogen atom involved in a migration process and the collective, detailed, quantitative results offer a stringent test for molecular dynamics calculations.

The revised and extended section now reads:

Opportunities for theoretical comparisons

The probability that an initial hydrogen site of an ethanol dication would contribute to the formation of a particular hydrogen-rich fragment was measured by combining the results from coincidence ion momentum imaging measurements of seven deuterium-tagged isotopologues of ethanol. These quantitative results provide insights into previously unresolved questions surrounding hydrogen migration in ethanol. For example, $H_{\alpha}H_{\alpha}H_{O}^{+}$ formation is far more likely than $H_{\alpha}H_{\alpha}H_{\beta}^{+}$, indicating that the roaming $H_{\alpha}H_{\alpha}$ [2] tends to sample the hydroxyl side of the molecule more than the methyl side. Hydrogen scrambling occurs in several channels, including H_2O^{+} formation, and hydronium ions predominantly form through migration of independent hydrogen atoms to the hydroxyl group, as suggested by Kling and coworkers [23]. Furthermore, the significant differences observed in the measured site-specific probabilities associated with complete and incomplete breakup channels calls for caution when generalizing measurements limited to complete breakup channels to the overall picture of the molecular fragmentation. The key to reaching these conclusions is the quantitative determination of the probability associated with hydrogen migration from each site.

By eliminating experimental ambiguities about the initial site of the hydrogen atoms participating in the hydrogen migration process, the data presented above put stringent constraints on theoretical modeling of intramolecular hydrogen dynamics. Molecular dynamics calculations of the fragmentation of the ethanol dication can now be benchmarked against experimentally determined probabilities for initial-site compositions yielding the hydrogen-rich molecular fragments. Additional composition-specific characteristics of the molecular fragments, such as KER or angular distributions, can also be extracted from the data and provide more detailed tests for future calculations of the underlying dynamics. The method can be extended to more complex molecular ions, and the anticipated collaborative efforts between experiment and theory should enhance the fundamental understanding of the coupling between electronic and nuclear dynamics in polyatomic molecular ions and, in turn, be applied to neutral molecules.

Reviewer #2 (Remarks to the Author):

The manuscript reports on experiments investigating the strong-field induced dissociation of ethanol. An experimental procedure is described for determining the relative probabilities of the formation of fragment ions containing hydrogen, where the hydrogen atoms start at different positions in ethanol. The key idea is to collect data for all types of deuterated molecules, and to determine the probabilities of the different pathways by an elaborate fitting procedure.

I think the manuscript is a very interesting one which would be well suited for publication in Nature Communications. The method should have a large impact on the field of strong-field chemistry and molecular physics.

Provided that the authors can reply to my comments below, I would recommend the manuscript for publication.

We thank Reviewer #2 for the positive comments about the manuscript. We reply to the comments below.

1. In Fig. 2, the branching ratios of ethanol and completely deuterated ethanol are compared. The difference is not large, so we can rule out quantum tunneling as the dissociation mechanism. Would it be possible to discuss quantitatively the difference in the branching ratios by referring to the mass difference? The difference could be a factor of $1/\sqrt{2}$?

The small magnitude of the isotopic differences and their impact on the site-specific probabilities, which we evaluated, are discussed in detail in a new SI section (Sec. 13). Please, see also our response to Reviewer #1, comments 5, and 6, which also involve the isotopic differences.

The main two causes for isotopic differences between H and D isotopologues are expected to be the different extent of the nuclear wavefunctions that, in turn, may lead to different Franck-Condon factors for similar electronic transitions, or a consequence of differences in the migration speed, like in the case of roaming. Both of these depend on the mass difference, but in a more complex way than a $1/\sqrt{2}$ factor. As this was not the aim of our work, quantitative understanding of the intriguing (small) isotopic differences, and their dependence on the mass, is left for future work.

2. In the introduction and in the discussion of H_3^+ formation on page 5. I think that the paper [J. Chem. Phys. 139, 181103 (2013) <https://doi.org/10.1063/1.4830397>] should be mentioned in the discussion.

This article, by K. Nakai, T. Kato, H. Kono, and K. Yamanouchi is one of several articles that have discussed H_3^+ production following ionization of a small polyatomic molecule, including ethane, methanol, ethanol, and a few larger molecules. We added this theoretical reference to that list as requested, as well as the underlying experimental study reported in T. Okino, Y. Furukawa, P. Liu, T. Ichikawa, R. Itakura, K. Hoshina, K. Yamanouchi, and H. Nakano, Chem. Phys. Lett. 419, 223 (2006).

The references to Nakai et al. [3] and Okino et al. [63], are now cited in the opening sentence of the paper

“The migration of hydrogen within polyatomic molecules, either collectively [1–3] or as a single atom or ion at a time [4–10], is a common process that impacts diverse applications such as

enzyme operation [11, 12], large scale studies of proteins [13], combustion [14–16], and atmospheric chemistry [17].”

the second paragraph of the introduction,

“Photo-induced intramolecular dynamics are usually explored theoretically by combining high-level electronic structure calculations with molecular dynamics methods (e.g. Refs. [1, 3, 9, 10, 23, 25]).”

and in the discussion of the H_3^+ results on page 6.

“Formation of H_3^+ is the most often studied of the molecular ions mentioned above [1–3, 7, 19, 22, 24–27, 44, 52, 63].”

3. I would appreciate if the “violin plots” could be explained briefly. Even if this type of plots is explained in [50], it would be helpful for the reader if a brief explanation could be provided to make the manuscript self-contained.

Point well taken. To help readers unfamiliar with violin plots we added to the caption of Fig. 3, more details on what is shown in these plots, and improved the clarity of that part of the caption, which now reads: “Relative probabilities of different initial-site compositions for complete fragmentation channels of ethanol dications. The left panel of each plot shows the probability distribution evaluated by a Monte-Carlo uncertainty propagation as a violin plot, where the white circle is the median, the black rectangle indicates the middle 50% of the distribution, while the thin black line indicates the range of the distribution excluding outliers, and the (vertical) width of the distribution denotes the probability of each value (see Ref. [59] for more details about violin plots).”

In addition, a more complete description of the violin plots has been added to Sec. 8 of the SI, including the source definition of the kernel density function. See the paragraph starting with “Since the relative probability distributions shown in Fig. S4...”

4. As I understand the manuscript, the experimental method and analysis is based on the comparison of data obtained by different isotopologues, and the assumption that the branching ratios for the different channels are the same for different isotopologues. However, the data in Fig. 2 shows that the branching ratios are not the same. Is the difference of the branching ratios somehow taken into account in the estimation of the error bars, or the shape of the violin plots?

That is correct, our approach was indeed to try to characterize the differences that occur due to the use of different isotopologues rather than correcting for these differences. Based on the extensive tests, discussed in detail in the new Sec 13 of the SI, we concluded that the range of values obtained from the Monte Carlo error propagation does a reasonable job of characterizing the spread in the site-specific probabilities due to the isotopic differences in the measured branching ratios. Additional discussion may be found in our response to Reviewer #1 above.

Minor remark:

1. In the abstract, an “H_alpha H_alpha” is mentioned, but this notation is defined later on in the paper, and cannot be understood only by reading the abstract. A description which can be understood on its own would be preferred.

This has been corrected in the revision of the abstract by referring to the $H\alpha$ atoms as the “central two hydrogen atoms”.

Reviewer #3 (Remarks to the Author):

We thank Reviewer #3 for the remarks below. Several of these comments align with comments made by the other Reviewers, highlighting the areas where our original messaging could be improved. This careful and constructive review of the manuscript helped us refine the resubmitted article considerably.

In this manuscript by Severt et al., the authors determined experimentally the originating sites of hydrogen in the formation process of tri-hydrogen cation from strong field induced ethanol dications. In this study, the authors used seven isotopologues of ethanol with deuterium and hydrogen located at different sites. With PIPICO measurements carried out by a COLTRIMS apparatus, branching ratios of many dissociation pathways were identified and a fitting process was adopted to extract the hydrogen initial site information. This is an interesting work and add to a large body of existing knowledge on dissociative double ionization of various small molecules. The employed method is not new because isotopologues of ethane have been previously used to extract similar information (see ref. 45 and also in JCP 134, 064324 (2011)). The current work deals with a larger molecule and more complicated dissociation pathways. But the results seem similar, i. e. that hydrogen scrambling is a significant process in strong field induced dissociation processed. The work is surely publishable, but I have no strong opinions regarding whether it is suitable for Nature Communications. I see it can go both ways. But I do have some comments for the authors to consider during a revision, if offered.

We agree with the reviewer assessment that deuterium tagging has been applied previously to determine the role of one site out of two, and we modified the introduction to make that clearer, including adding the suggested reference (Ref. [41]). The challenge, however, is to generalize this approach beyond two hydrogen sites, since there are only two hydrogen isotopes are readily available. We have demonstrated this by determining the site-specific probabilities for hydrogen migration forming a few hydrogen-rich fragments upon the fragmentation of ethanol dications. Furthermore, we show that the method is general, i.e., it can be extended to more complex molecules. This is mainly described in the modified text (page 1) reading:

“...Of special relevance to the present work are mass-spectrometry and ion-imaging studies in which quantitative comparisons of different tri-hydrogen (H_3^+) formation processes are obtained in ethane [41, 42] and methanol [1, 43, 44]. Extending this work beyond molecules with two nonequivalent hydrogen sites is a challenge since there are only two readily available isotopes of hydrogen, and thus no single deuterium-tagged experiment can identify the role of all hydrogen sites. The goal of our study is to employ deuterium tagging and determine the contribution of each hydrogen site in larger molecules, which display compelling hydrogen migration dynamics, and enable quantitative comparisons between experiment and theory. Ethanol, with three hydrogen sites, has attracted considerable experimental and theoretical interest due to the complexity of its dissociation [2, 7, 23, 25–27, 45–48] and thus makes an excellent prototype system for this study.”

Hopefully, these text modifications more appropriately highlight the significance of the method we have demonstrated and the impact this benchmark measurement may have on advancing molecular dynamics theory.

In addition, the data presented here confirm conclusions or answer questions left unsettled in previous high visibility publications on this topic [2,23].

1. This is an experimental work. But the first sentence of the abstract gives the impression that it might include theory work, which is not true. I think the author should rewrite or remove that sentence.

Reviewer #1 also expressed similar comments, except regarding the introduction rather than the abstract. While removing experimental ambiguities to facilitate a more stringent benchmark for theory is a primary motivation for this work, we regret that our attempts to frame the introduction and the abstract in that way were easily misinterpreted. We did not intend to suggest we did any calculations. We have extensively rewritten both the abstract and the introduction to clarify this distinction.

2. The C-C bond breaking channel was not discussed. This is a main channel and will reveal the degree of hydrogen scrambling. The authors should include it in Fig. 2 also.

The main analysis presented in the paper and, in particular, illustrated in Fig. 2, is restricted to “complete” two-body breakup channels of the ethanol dication involving tri-hydrogen, water, hydronium or methane ion formation. While the case of methane production does involve the C-C bond breaking (and is already included), for the three other ions the C-C bond breaking would mean that the molecule breaks up in at least three fragments. Therefore, within the framework of the present manuscript, we cannot directly add further channels involving C-C bond breaking to Fig. 2. To address the referee’s request in a different way, we added four channels with C-C bond breaking to the new Fig. S9 and incorporated those channels into the analysis of Sec. 13 of the SI.

The breaking of the C-C bond occurs frequently, and some of those channels involve hydrogen scrambling. We are currently exploring some of these channels, in particular the seven double-ionization channels associated with both H_3^+ formation and the breaking of the C-C bond. There are also channels that could form H_2O^+ , H_3O^+ , or CH_4^+ in addition to breaking a C-C bond, although extracting all these channels for the heavier fragments is more difficult than for H_3^+ due to the density of the channels in that region of the PIPICO spectrum.

Distilling these channels down to a single C-C bond breaking channel that could be added to Fig. 2 does not seem feasible to us. Introducing the discussions of all the above dissociation channels into the current manuscript would, in our view, make the resulting narrative unwieldy. We anticipate the need for a follow-up publication to fully explore these topics.

We also modified the titles of Figs. 2-4 to make it explicit that we are only presenting the results for certain dissociation channels.

3. There is serious congestion in the PIPICO spectrum. The authors explained they used momentum conservation to select intended channels. How robust are the branching ratios when the momentum conservation criterion is varied (sum $P < ?$).

As the reviewer suggests with this comment, the gates on the PIPICO spectrum are more restrictive than the momentum conservation criteria. We typically require that $\sum p_i < 30$ atomic

units in the two-body channels, and the results are relatively insensitive to changing this value moderately. The random coefficient determination and the construction of the gates on the PIPICO spectrum, both of which are determined by the person doing the analysis, have a larger effect. These two effects were included in the analysis of the systematic uncertainty, as discussed in SI Sec. 8.

The PIPICO spectra is indeed fairly congested in the views presented in the SI. We have added two expanded views to figure S3 (panels c and d) to give a better sense of the TOF resolution (~ 0.25 ns, binned to 1 ns) and separation between the channels. In cases where channels do cross each other, there are additional steps that can be employed, such as using reflection to evaluate half the coincidence channel instead of the entire range. When these additional methods are employed, the uncertainty is increased appropriately.

4. The authors should cite JCP 134, 064324 (2011).

As noted above, we have now cited this reference as Ref. [41]. It appears in the introduction,

“Of special relevance to the present work are mass-spectrometry and ion-imaging studies in which quantitative comparisons of different tri-hydrogen (H_3^+) formation processes are obtained in ethane [41, 42] and methanol [1, 43, 44].”

on the left column of page 2,

“Characterization of all of the possible resulting ion compositions shows that seemingly unlikely processes, such as hydrogen scrambling [41, 42, 53, 54], are sometimes significant.”

The left column of page 3,

“As with many deuterium substitution techniques, the fundamental assumption is that the mass difference between $1H$ and $2D$ does not significantly affect the properties to be studied. This common assumption [41, 42, 55, 56] was verified in our case by comparing the branching ratios from all-hydrogen isotopologue, #1 in Fig. 1(c), to the analogous channels from the all-deuterium isotopologue, #7 in Fig. 1(c)”.

and in the Methods section,

By avoiding channels with mass overlaps, we avoid having to make any assumptions that different hydrogen sites behave in the same way, as are sometimes invoked in other cases [41, 42].

In closing, we thank all the reviewers for their careful reading of the manuscript and constructive comments. We believe the resubmitted version of the manuscript addresses all of the reviewer’s critiques and we look forward to publication of this work.

REVIEWERS' COMMENTS

Reviewer #1 (Remarks to the Author):

I would like to thank the authors for addressing the points that were raised by the reviewers so thoroughly. The motivation for the research, the work carried out by the authors, and their interpretation of the results are all explained much more clearly in the revised manuscript.

Reviewer #2 (Remarks to the Author):

I believe that the authors have responded in an appropriate way to the comments by me and the other reviewer. The manuscript is now very clear.
I think the results presented are a very interesting contribution to strong-field induced dissociation of multi-atom molecules. I recommend publication of the manuscript in its present form.

Reviewer #3 (Remarks to the Author):

The authors has adequately addressed my concerns. I support the publication of this manuscript.